# Taming Discretised PDDL+ through Multiple Discretisations

**Primary Keywords:** *Knowledge Representation/Engineering*

## Abstract

The PDDL+ formalism allows the use of planning techniques in applications that require the ability to perform hybrid discrete-continuous reasoning. PDDL+ problems are notoriously challenging to tackle, and to reason upon them a well-established approach is discretisation. Existing systems rely on a single discretisation delta or, at most, two: a simulation delta to model the dynamics of the environment, and a planning delta, that is used to specify when decisions can be taken. However, there exist cases where this rigid schema is not ideal, for instance when agents with very different speeds need to cooperate or interact in a shared environment, and a more flexible approach that can accommodate more deltas is necessary. To address the needs of this class of hybrid planning problems, in this paper we introduce a reformulation approach that allows the encapsulation of different levels of discretisation in PDDL+ models, hence allowing any domain-independent planning engine to reap the benefits. Further, we provide the community with a new set of benchmarks that highlights the limits of fixed discretisation.

## Introduction

The ability to represent hybrid discrete-continuous changes is crucial to exploit automated planning techniques in real-world applications. The PDDL+ language has been introduced and designed to support a compact encoding of models involving hybrid changes, via the use of specialised constructs such as events and processes (Fox and Long 2006).

Hybrid PDDL+ problems are notoriously challenging to tackle, due to the intertwined nature of numeric variables and time. A well-established approach to reason upon hybrid PDDL+ problems is discretisation (Della Penna, Magazzeni, and Mercorio 2012; Percassi, Scala, and Vallati 2023b), which allows breaking down complexity by assuming the time is discrete, and so are the actual numeric changes. A similar assumption is also made in the simpler context of temporal planning through durative actions (Dvorak et al. 2014; Rintanen 2015; Cushing et al. 2007). An important aspect of this approach is the ability to re-use well-established and general search techniques based on forward state-based exploration to tackle PDDL+ problems; it is indeed widely exploited by existing domain-independent planning engines

such as DINO (Piotrowski et al. 2016), UPMURPHI (Della Penna, Magazzeni, and Mercorio 2012) and ENHSP (Scala et al. 2016). The first two solvers rely on only one discretisation step for both simulation (process and events) and decision (actions), while ENHSP utilises a more advanced approach that can support two deltas: a (usually smaller) simulation delta to approximate complex dynamics and a (usually larger) planning delta to reduce the burden on planning by avoiding unnecessary decision points.

Notably, there can be cases where even the advanced technique of using two different discretisation deltas does not allow to efficiently reason upon the dynamics of the problem at hand. In the logistics context, for example, it is common to have a range of means of transport, each having a different speed and a different granularity of timings in which actions must be performed (e.g. a plane is faster than a truck which is faster than a delivery man) and if the different agents involved have to coordinate, they must necessarily do it at the discretisation step of the slower one, making the solving unnecessarily challenging. Even the same agent could benefit from different granularity in different moments of the plan: for example, a ship must be finely controlled while manoeuvring in the harbour, but its course can be sporadically altered while at open sea. To effectively address the described class of hybrid problems, approaches capable of supporting even more than two discretisation deltas are needed.

In this work, we address this need by introducing a reformulation approach that encapsulates such multiple deltas straight into the PDDL+ models, hence allowing domain-independent planning engines to exploit the benefits. More precisely, we formally define the dynamic planning-discretised PDDL+ problem, and introduce a sound and complete compilation allowing one to generate a corresponding PDDL+ model that encodes the notion of multiple deltas. Any planning engine that supports PDDL+ can reason upon the reformulated models, thus extending the ability of existing systems to solve challenging hybrid problems. Further, we present an innovative PDDL+ benchmark domain, based on a realistic example, that stresses the need for reasoning with multiple deltas. Our extensive empirical analysis demonstrates the merits of the introduced reformulation on a range of PDDL+ benchmarks and planning approaches.

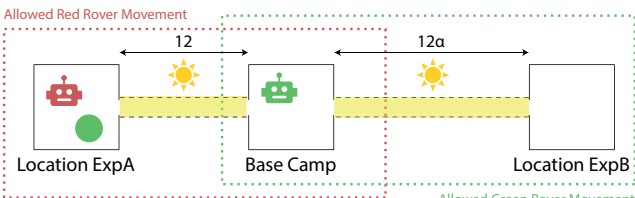
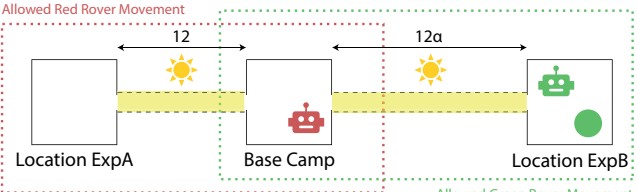

Figure 1: A representation of the initial state and goal condition of the COOPROVERS motivating example.

## Related Work

The PDDL+ formalism is the most expressive formalism of the PDDL family of languages, that also includes PDDL (McDermott et al. 1998) and PDDL2.1 (Fox and Long 2003). The modelling capabilities of PDDL+ have enabled the use of automated planning to solve complex real-world problems such as traffic control (Vallati et al. 2016), safety requirements for cyber-physical systems (Aineto et al. 2023), train dispatching (Cardellini et al. 2021), unmanned aerial vehicle control (Kiam et al. 2020), pharmacokinetic optimisation (Alaboud and Coles 2019), and popular video games (Piotrowski et al. 2023). Due to its expressive power, finding solutions for PDDL+ problems remains a daunting challenge. This is further compounded by the scarcity of planners capable of effectively handling PDDL+ problems.

Existing solvers deal with the continuous nature of Ordinary Differential Equations (ODEs) in two ways: either by (i) solving the underlying integral, or (ii) by discretising the time horizon and treating the ODEs as discrete sums. In the first category, solvers like SMTPLAN+ (Cashmore, Magazzeni, and Zehtabi 2020) and dReal (Bryce et al. 2015) make use of SMT techniques and mathematical solvers to solve the integrals. Due to the mathematical complexity of these operations, both planners apply restrictions to the possible set of functions which can be expressed in the ODEs. To the second category, instead, belongs planners like UPMUR-PHI (Della Penna, Magazzeni, and Mercorio 2012), DiNo (Piotrowski et al. 2016) and ENHSP (Scala et al. 2016) which make use of the Discretise & Validate approach (Della Penna, Magazzeni, and Mercorio 2012), allowing them to deal with a larger set of ODEs functions but being dependant of the discretisation step for the validation of the plan.

Both DiNo and UpMurphi consider a single delta, that is used for both simulating the evolution of the dynamic environment and for identifying decision points for planning. A more advanced approach, presented by (Ramirez et al. 2017) and supported by ENHSP, is to consider two different deltas: a simulation delta, to be as small as possible to better approximate complex hybrid dynamics, and a planning delta, that can be discretionally large, to reduce the burden on the planning process by avoiding decision points when no actions are likely to be applicable. An approach similar in nature, but domain-specific, has been proposed for the Train Dispatching Problem (Cardellini et al. 2021), where the ENHSP planner has been modified to skip irrelevant decisions points when controlling the dispatching of trains.

## Motivating Example

In this section, we present a novel domain, COOPROVERS, in which two agents operate at different speeds and need to coordinate to reach the stated goals. Figure 1 provides an example of an initial state (left) and a goal condition (right) in which two rovers (Red and Green) are performing two experiments (A and B) in two separate locations and need to exchange a tool. For safety reasons, the rovers are only allowed to move from the base camp to their working zone, and hence they can only meet at the base camp to exchange the mentioned tool. The two rovers are equipped with a battery and solar panels that allow them to recharge. Since the location of the Experiment B is $\alpha$ times more distant from the base camp than that of Experiment A, the Green rover has been equipped with a lighter, more efficient battery (discharges at $\frac{20}{\alpha}\%/m$), consuming less than the Red rover ($20\%/m$) and allowing longer trips. At any point, while moving between the locations, the rovers can deploy their solar panels and recharge (at the speed of $1\%/s$) for some time before restarting their trip. The battery must always stay above $20\%$ to allow emergency operations and the deployment of solar panels. The rovers are also equipped with a holding container for transporting tools. The speed of the two rovers is the same ($1m/s$) but, given the differences in distances to be covered and discharging rates, their movements should be modelled and controlled with different granularities. For example, with $\alpha = 100$, the Green rover would need to move 1.2 km at a velocity of 1 m/s, discharging at 0.2%/m. In this case, it is easy to notice that the Green robot would benefit more from a discretisation step one hundred times larger than the Red robot.

## Background

A PDDL+ planning problem, denoted by $\Pi$, is a tuple $\langle F, X, I, G, A, E, P \rangle$, where $F$ is a set of Boolean variables and $X$ is a set of numeric variables taking values from $\{\top, \bot\}$ and $\mathbb{Q}$, respectively. These variables can be used in propositional formulas with numeric and Boolean conditions. Numeric conditions are of the form $\langle \xi \bowtie 0 \rangle$, where $\xi$ is a numeric expression over $X$ and $\mathbb{Q}$, and $\bowtie \in \{\leq, <, =, >, \geq\}$. Boolean conditions are of the form $\langle f = b \rangle$ with $f \in F$ and $b \in \{\top, \bot\}$. A formula is therefore a propositional formula using standard connectives from logic involving numeric and Boolean conditions. $I$ is the description of the initial state, expressed as a full assignment to all variables in $X$ and $F$. $G$ is the description of the goal,

expressed as a formula. $A$ and $E$ are the sets of actions and events, respectively. An action or event is a pair $\langle p, e \rangle$, where $p$ is a formula and $e$ is a set of Boolean or numeric effects. A Boolean assignment has the form $\langle f := b \rangle$, where $f \in F$ and $b \in \{\bot, \top\}$. A numeric assignment has the form $\langle op, x, \xi \rangle$, where $op \in \{asgn, inc, dec\}$, $x \in X$, and $\xi$ is a numeric expression over $X$ and $\mathbb{Q}$. Specifically, $op$ can be the contraction of the keywords $assign$ ($x := \xi$), $increase$ ($x := x + \xi$) and $decrease$ ($x := x - \xi$). $P$ is a set of processes and a process is a pair $\langle p, e' \rangle$, where $p$ is a formula and $e'$ is a set of continuous numeric effects expressed as pairs $\langle x, \xi \rangle$, where $x \in X$ and $\xi$ is a numeric expression defined as above. $\xi$ represents the additive contribution to the first derivative of $x$ as time flows continuously. In the discrete context, $\xi$ is the additive contribution to the discrete change of $x$. Let $a = \langle p, e \rangle$ be an action, event, or process, we use $pre(a)$ to refer to the precondition $p$ of $a$, and $eff(a)$ to the effect $e$ of $a$. In the following, we will use $a$, $\rho$, and $\varepsilon$ to refer to a generic action, process, and event, respectively.

A PDDL+ plan $\pi_t$ is a pair $\langle \pi, t_e \rangle$, where $\pi = \langle \langle a_1, t_1 \rangle, ..., \langle a_n, t_n \rangle \rangle$ is a sequence of timestamped actions and $t_e \in \mathbb{Q}_{\geq 0}$ is the makespan within the plan $\pi$ is executed.

A state $s$ is a full assignment of the variables $X \cup F$. An action $a$ (event $\varepsilon$) is applicable (is triggered) in a state $s$ iff $s \models pre(a)$ ($s \models pre(\varepsilon)$). For describing how a state changes when an action (event) is executed (triggered) we use the transition function $\gamma(s, z)$. Given a state $s$ and an action/event $z \in A \cup E$, $\gamma(s, z)$ denotes the state resulting from the application of $a$ in $s$ accordingly to the effect $eff(z)$. The difference between actions and events is that the former prescribe *may transitions* under the control of the agent and can be executed if the current state meets the preconditions, while the latter prescribe *must transitions*, i.e., events are triggered immediately if their preconditions are met.

Following some early works (Shin and Davis 2005; Percassi, Scala, and Vallati 2023b), we formalise the PDDL+ discrete semantics through the notion of time points, histories and plan projections. Given a discretisation step $\delta \in \mathbb{Q}_{>0}$, a time point, denoted by $T$, is a pair $\langle t = \delta \cdot n, n' \rangle$, where $n, n' \in \mathbb{N}$; $t$ denotes the clock of $T$ while $n'$ is the counter used to order actions and events happening at $t$. Time points are ordered lexicographically. A history $\mathbb{H}$ over a sequence of time point $T_{\mathbb{H}}$ maps each element from $T_{\mathbb{H}}$ into a situation. A *situation at time point* $T$ is the tuple $\mathbb{H}(T) = \langle \mathbb{H}_A(T), \mathbb{H}_s(T) \rangle$, where $\mathbb{H}_A(T)$ is the action executed at time point $T$ (if any) and $\mathbb{H}_s(T)$ is the state associated with $T$. We denote by $\mathbb{H}_s(T)[v]$ and $\mathbb{H}_s(T)[\xi]$ the value assumed by $v \in F \cup X$ and by a numeric expression $\xi$, respectively, in state $\mathbb{H}_s(T)$. $E_{trigg}(T)$ indicates the sequence of events triggered in $T$. [1]

The validity of plans relies on defining the discrete PDDL+ plan projection, which describes how $\pi_t$ is projected onto a history, taking into account the effects of actions and changes yielded by events and processes. This projection is

---

constructed through two types of transitions: instantaneous and temporal. Instantaneous transitions originate from the execution (triggering) of actions (events), whereas temporal transitions result from the passage of a discrete quantum of time. Each transition is associated with a starting time point and one linked to the resulting state after the transition. These time points are referred to as significant (STPs).

Intuitively, we define the plan projection based on a set of rules that describe how history progresses over time. The first rule (R1) states that if an event is triggered at a specific time point, a successor state must exist with the same clock time and an increased counter. The second rule (R2) states the same for actions. The third and fourth rules (R3-R4) are used to ensure that the actions in a PDDL+ plan are projected, preserving their original ordering. The fifth rule (R5) is used to describe how numeric variables change over time in a discrete fashion when time advances by a discrete quantity. Notably, continuous numeric changes are discretised according to the formula $\Delta(\xi, \delta) = \delta \cdot \xi$.

**Definition 1** (Discrete PDDL+ Plan Projection). *Let $\Pi$ be a PDDL+ problem, $\delta \in \mathbb{Q}_{>0}$ a discretisation step, $\mathbb{H}$ an history and $\pi_t = \langle \pi, t_e \rangle$ a plan for $\Pi$. We say that $\mathbb{H}$ is a discrete projection of $\pi_t$ which starts in $I$ iff $\mathbb{H}$ induces the STPs $T_{\mathbb{H}} = \langle T_0 = \langle t_0 = 0, 0 \rangle, ..., T_m = \langle t_m = t_e, n_m \rangle \rangle$, where either $t_{i+1} = t_i + \delta$ or $t_{i+1} = t_i$, $\mathbb{H}_s(T_0) = I$ and, for all $i \in \{0, ..., m\}$, the following rules hold:*

**R1** [Instantaneous Transition (events)] $E_{trigg}(T_i) \neq \langle \rangle$ *iff* $\mathbb{H}_s(T_{i+1}) = \gamma(\mathbb{H}_s(T_i), E_{trigg}(T_i))$, $\mathbb{H}_A(T_i) = \langle \rangle$, $t_{i+1} = t_i$ *and* $n_{i+1} = n_i + 1$;

**R2** [Instantaneous Transition (actions)] $\mathbb{H}_A(T_i) \neq \langle \rangle$ *iff* $\mathbb{H}_s(T_{i+1}) = \gamma(\mathbb{H}_s(T_i), \mathbb{H}_A(T_i))$, $E_{trigg}(T_i) = \langle \rangle$, $t_{i+1} = t_i$ *and* $n_{i+1} = n_i + 1$;

**R3** [Action Projection] $\langle a_i, t_i \rangle \in \pi$ *iff it exists one and only one* $T_i = \langle t', n \rangle \in T_{\mathbb{H}}$ *s.t* $\mathbb{H}_A(T_i) = \langle a_i \rangle$ *and* $t' = t_i$;

**R4** [Actions Ordering] *for each* $\langle a_i, t_i \rangle, \langle a_j, t_j \rangle$ *in* $\pi$, *with* $i < j$ *and* $t_i = t_j$ *there exists* $T_k, T_z$ *in* $T_{\mathbb{H}}$ *such that* $\mathbb{H}_A(T_k) = \langle a_i \rangle$ *and* $\mathbb{H}_A(T_z) = \langle a_j \rangle$ *and where* $t_k = t_z = t_i$ *and* $n_k < n_z$;

**R5** [Temporal Transition] *for each pair of contiguous STPs* $T_i = \langle t_i, n_i \rangle$, $T_{i+1} = \langle t_{i+1}, n_{i+1} \rangle$ *such that* $t_{i+1} = t_i + \delta$, *we have that* $n_{i+1} = 0$ *and the value of each numeric variable $x \in X$ is updated as:*

$$\mathbb{H}_s(T_{i+1})[x] = \mathbb{H}_s(T_i)[x] + \sum_{\substack{\langle x', \xi \rangle \in eff(\rho), \ x' = x \\ \rho \in P \ s.t. \ \mathbb{H}_s(T_i) \models pre(\rho)}} \mathbb{H}_s(T_i)[\Delta(\xi, \delta)]$$

*and values of unaffected variables remain unchanged (frame-axiom).*

**Definition 2** (Valid PDDL+ plan under $\delta$ discretisation). *$\pi_t$ is a valid plan for $\Pi$ under $\delta$ discretisation iff $\mathbb{H}_s(T_m) \models G$ and, for each $T \in T_{\mathbb{H}}$ such that $\mathbb{H}_A(T) = \langle a \rangle$, then $\mathbb{H}_s(T) \models pre(a)$.*

**Motivating example (cont'd).** We are now in the position to illustrate how to model[2] the COOPROVERS domain using PDDL+. The movement of a rover $r$ from

---

[1] We assume PDDL+ problems are event-deterministic, i.e., given a state where multiple events are triggered together, we can sequence them arbitrarily, always obtaining the same outcome (Fox and Long 2006).

[2] The full PDDL+ formulation is available at https://anonymous.4open.science/r/deltaExperiments-D585.

two connected locations `a` and `b` is managed through the triplet of action `startMoving(r,a,b)`, process `moving(r,a,b)`, and event `endMovement(r,a,b)`. The `moving(r,a,b)` action is active only when the battery is above the threshold of $20\%$ and keeps updating a variable `dRun(r,a,b)` whose role is to track the progress of the rover in going from `a` to `b`. During the movement, process `discharge(r)` models the draining of the battery, and does so with a rate of `cRate(r)`. The planning engine can decide to interrupt and restart the movement through action `startCharging(r)` and action `stopCharging(r)`, respectively. Between these two actions, the process `charging(r)` gets activated, and the rover battery charges with a rate of $1\%/s$. To collect and exchange tools, the actions `drop(r,o)` and `pick(r,o)` model the handling of the object `o` by rover `r`. In the initial condition, the robots are set in the configuration shown in Figure 1 (left). The goal is to reach a state where the tool has been brought to the location of Experiment B (Figure 1 (right)).

## Dynamic Planning-Discretised PDDL+

To address the kind of hybrid problems that require the ability to deal with different dynamics, here we characterise the dynamic planning-discretised PDDL+ problem.

A dynamic planning-discretised PDDL+ problem (shortened as PDDL$\delta$+ problem) is the tuple $\langle \Pi, K_\delta = \langle J, \nabla \rangle \rangle$, where $\Pi$ is a PDDL+ problem defined as above and $K_\delta$ is the *discretisation knowledge* detailed as follows. $J$ is a function $A \cup E \to \{1, ..., m\}$ which partitions the set of actions and events in $m$ classes such that $A = \bigcup_{j=1}^{m} A_j$ and $E = \bigcup_{i=j}^{m} E_j$, where $A_j = \{a \in A \mid J(a) = j\}$ and $E_j = \{\varepsilon \in E \mid J(\varepsilon) = j\}$. The number of partitions induced by $J$ defines the number of discretisation variables, i.e., $\delta^m = \{\delta_1, ..., \delta_m\}$, with each of them taking values in $\mathbb{Q}_{>0}$. Intuitively, every $\delta_j$ manages a different aspect of the problem by controlling when actions from $A_j$ can be executed. $\nabla$ is the function which controls the dynamic of the discretisation steps, that is, how the $\delta^m$ variables change according to the actions applied and the triggered events. Such a function maps every action and event into a positive rational number plus a special symbol $\kappa$, i.e., $\nabla : A \cup E \to \mathbb{Q}_{>0} \cup \{\kappa\}$; the special symbol $\kappa$ is the *persist* value, and it represents that the affected discretisation variable remains unchanged when an action (event) $z$ with $\nabla(z) = \kappa$ is applied (triggered). With a little abuse of notation, we allow the $\nabla$ function to also accept the initial state as input and return a full assignment of $\delta_m$, i.e., $\nabla(I) = \{\langle \delta_i := \delta_i^0 \rangle \mid \delta_i \in \delta^m\}$. This allows us to initialise the discretisation variables in the initial state.

A discretisation knowledge $K_\delta$ may induce a non-deterministic behaviour w.r.t. events. In particular, it is known that events can generate non-determinism in PDDL+ problems (Fox and Long 2006) and this can also affect the discretisation variables $\delta^m$. That said, we define a discretisation knowledge $K_\delta$ as *event-deterministic* iff for each state $s$, and for all $\varepsilon, \varepsilon' \in E$ where $J(\varepsilon) = J(\varepsilon')$ and $s \models pre(\varepsilon) \wedge pre(\varepsilon')$, it holds that $\nabla(\varepsilon) = \nabla(\varepsilon')$. In simpler

terms, for any pair of events belonging to the same partition that can be triggered simultaneously, the $\nabla$ function consistently prescribes the same discretisation value.

Intuitively, solving a PDDL$\delta$+ problem $\langle \Pi, K_\delta \rangle$ consists in finding a valid PDDL+ plan for $\Pi$ such that every executed action is compatible with the discretisation steps prescribed by $K_\delta$.

To formally define the semantics of PDDL$\delta$+, we begin by introducing a new set of $m$ memory variables denoted as $\{M_j = \langle \hat{t}_j, \delta_j \rangle \mid j \in \{1, ..., m\}\}$. Each element in this set is a pair of positive rational numbers. Essentially, for a given partition $j \in \{1, ..., m\}$, the first component of $M_j$, i.e., $\hat{t}_j$, represents the most recent timestamp at which an action (event) from $A_j$ ($E_j$) was executed (triggered). The second component, i.e., $\delta_j$, indicates the latest discretization step assigned to partition $j$ based on the $\nabla$ function. The combination of these two elements determines all the following timestamps in which the actions from each partition are applicable.

We now extend the definition of a history to keep track of the memory variables introduced so far. To be specific, given a history $\mathbb{H}$, $\mathbb{H}_K(T)$ specifies a full assignment to the memory variables at time point $T$. We denote by $\mathbb{H}_K(T)[M_j]$ the value assumed by $M_j = \langle \hat{t}_j, \delta_j \rangle$ at $T$. In the plan projection, for the first STP, such an assignment is equal to $\mathbb{H}_K(T_0) = \{\langle \hat{t}_j = 0, \delta_j = \delta_j^0 \rangle \mid j \in \{1, ..., m\}\}$. Furthermore, $\mathbb{H}_K$ is updated whenever an action is executed or an event is triggered, while it persists when time flows. Whenever an action $a$ from partition $j$ ($a \in A_j$) is applied at time $t_i$ (R2 applies), the variable $\hat{t}_j$ is updated to $t_i$ to keep track of the most recent timestamp when an action from $A_j$ was executed. Simultaneously, the discretisation variable $\delta_j$ is updated based on $\nabla(a)$: if $\nabla(a)$ is equal to $\kappa$ (persist value), the current discretisation value is retained; otherwise, it is modified. When an event $\varepsilon$ from partition $j$ ($\varepsilon \in E_j$) is triggered at time $t_i$ (R1 is applied), the variables $\hat{t}_j$ and $\delta_j$ are updated to $t_i$ and $\nabla(\varepsilon)$, respectively, only if $\nabla(\varepsilon) \neq \kappa$. Otherwise, the current values of $M_j$ are retained. Definition 1 is extended by reshaping R1-R2, which are responsible for handling actions and events.

**Definition 3** (Discrete PDDL$\delta$+ Plan Projection). *The discrete PDDL$\delta$+ plan projection of plan $\pi_t$ is defined in the same way as a discrete PDDL+ plan projection, except for R1 and R2 which are extended as follows:*

**R1** $E_{trigg}(T_i) \neq \langle \rangle$ *iff* $\mathbb{H}_s(T_{i+1}) = \gamma(\mathbb{H}_s(T_i), E_{trigg}(T_i))$, $\mathbb{H}_A(T_i) = \langle \rangle$, $t_{i+1} = t_i$ *and* $n_{i+1} = n_i + 1$; *furthermore, given* $\varepsilon$ *in* $E_{trigg}(T_i)$, $\mathbb{H}_K(T_{i+1})[M_{J(\varepsilon)}] = \langle \mathcal{U}_T(\varepsilon), \mathcal{U}_D(\varepsilon) \rangle$;

**R2** $\mathbb{H}_A(T_i) = \langle a \rangle$ *iff* $\mathbb{H}_s(T_{i+1}) = \gamma(\mathbb{H}_s(T_i), \mathbb{H}_A(T_i))$, $E_{trigg}(T_i)) = \langle \rangle$, $t_{i+1} = t_i$ *and* $n_{i+1} = n_i + 1$; *furthermore,* $\mathbb{H}_K(T_{i+1})[M_{J(a)}] = \langle t_i, \mathcal{U}_D(a) \rangle$.

*The functions used for updating $\mathbb{H}_K(T_{i+1})$ are defined as:*

$$\mathcal{U}_T(z) = \begin{cases} t_i & \text{if } \nabla(z) \neq \kappa \\ t_{last} & \text{otherwise} \end{cases}$$

$$\mathcal{U}_D(z) = \begin{cases} \nabla(z) & \text{if } \nabla(z) \neq \kappa \\ \delta_{last} & \text{otherwise} \end{cases}$$

| | $m = 1$ | $m > 1$ |
|---|---|---|
| $\nabla$ is globally flat | *Unitary-Static* | *Multiple-Static* |
| $\nabla$ is not globally flat | *Unitary-Dynamic* | *Multiple-Dynamic* |

Table 1: Different levels of discretisation control allowed by the discussed PDDL$\delta$+ framework.

*where* $\langle t_{last}, \delta_{last} \rangle = \mathbb{H}_K(T_i)[M_{J(z)}]$ *and* $z \in A \cup E$.

There is also the need to extend Definition 2 for the PDDL$\delta$+ plan validity. In particular, a plan $\pi_t$ is valid for a PDDL$\delta$+ problem $\langle \Pi, K_\delta \rangle$ iff $\pi_t$ is valid for $\Pi$ and every action of $\pi_t$ is executed in a time-stamp compatible with $K_\delta$.

**Definition 4.** *A* PDDL+ *plan* $\pi_t$ *is valid for a* PDDL$\delta$+ *problem* $\langle \Pi, K_\delta \rangle$ *iff* $\pi_t$ *is valid for* $\Pi$ *and for each* $T = \langle t, n \rangle \in T_{\mathbb{H}}$ *such that* $\mathbb{H}_A(T) = \langle a \rangle$, *there exists an* $s \in \mathbb{N}$ *such that* $t = t_{last} + s \cdot \delta_{last}$, *where* $\langle t_{last}, \delta_{last} \rangle = \mathbb{H}_K(T)[M_{J(a)}]$.

## Levels of Discretisation Control

Different levels of discretisation control can be achieved based on the definition of $K_\delta$, formalised as follows. In particular, we consider two dimensions: the number of partitions of $A \cup E$ induced by $J$, i.e., $m$, and the dynamic of $\nabla$ for each partition. If $m = 1$, $K_\delta$ induces a *unitary* PDDL$\delta$+ problem while, if $m > 1$, a *multiple* one. If the function $\nabla(z) = \kappa$ for each $z \in A_j \cup E_j$ we say that $\nabla$ is *flat* w.r.t. the partition $j$, otherwise is not. When $\nabla$ is flat for each partition (*globally flat*), we say that $K_\delta$ induces a *static* PDDL$\delta$+ problem, otherwise a *dynamic* one. Table 1 shows the different levels of discretisation control that can be achieved using the PDDL$\delta$+ framework.

Most of the PDDL+ discrete planning engines leverage a unique discretisation step $\delta_e$, both to model the granularity of the environmental changes and the agent's actions. Such a model can be expressed within the PDDL$\delta$+ framework by a discretisation knowledge $K_\delta$ in which the function $J$ induces a single partition so that, $J(z) = 1$ for each $z \in A \cup E$, there is a single discretisation step $\delta^1 = \{\delta_1\}$ that is initialised as $\{\langle \delta_1 := \delta_e \rangle\}$ and finally $\nabla(z) = \kappa$ for each $z \in A \cup E$.

ENHSP goes one step further in the direction of handling PDDL+ models with multiple discretisation steps. It separates the discretisation step for controlling the granularity of environmental changes $\delta_e$ and the one for controlling the granularity of the agent's actions $\delta_p$. Such a model is useful when it is necessary to have a fine approximation of the environmental dynamics while the agent, being characterised by a slower dynamic, performs actions more sporadically. Such a model can be expressed within the PDDL$\delta$+ framework by a $K_\delta$ in which the function $J$ induces a single partition so that, $J(z) = 1$ for each $z \in A \cup E$, there is a single discretisation step $\delta^1 = \{\delta_1\}$ that is initialised as $\{\langle \delta_1 := \delta_p \rangle\}$ and finally $\nabla(z) = \kappa$ for each $z \in A \cup E$.

We have shown that the discretisation models currently supported by the PDDL+ discrete planning engines fall in the *Unitary-Static* level of our framework. All models outside this are not supported by existing PDDL+ reasoners.

**Motivating example (cont'd).** We now show how the discretisation knowledge $K_\delta$ can be expressed in the COOPROVERS domain. Intuitively, the actions and events can be partitioned by the rover which performs the action or is subject to the events. For example, $J(\texttt{startCharging(red)})$ and $J(\texttt{startMoving(red,expA,bc)})$ are set equal to 1 and $J(\texttt{startCharging(green)})$ and $J(\texttt{startMoving(green,expB,bc)})$ equal to 2. This partition induces the set $\delta^2 = \{\delta_1, \delta_2\}$. The function $\nabla$ is set to allow for (i) differentiating the various time scales of the two rovers when they are moving, and (ii) allowing for the same timescale when the two rovers are charging. For this reason, $\nabla(\texttt{startMoving(red,expA,bc)})$ (and the symmetric action for moving from bc to expA) is set to 3 while $\nabla(\texttt{startMoving(green,expB,bc)})$ (and symmetric) is set to $3\alpha$, allowing for (i). $\nabla(\texttt{startCharging(red)})$ and $\nabla(\texttt{startCharging(green)})$ are all set to 30, allowing for (ii). For all the other actions and events, $\nabla$ returns $\kappa$. The initial condition sets the initial deltas to their respective delta of movements: $\nabla(I) = \{\langle \delta_1 := 3 \rangle, \langle \delta_2 := 3\alpha \rangle\}$. Since $m = 2$ and the $\nabla$ function is not flat, the motivating example falls in the *Multiple-Dynamic* discretisation control level, i.e., the most general among the introduced levels.

## Encoding of $K_\delta$ in PDDL+

Let $\langle \Pi = \langle F, X, I, G, A, E, P \rangle, K_\delta = \langle J, \nabla \rangle \rangle$ be a PDDL$\delta$+ problem. We introduce the FLAT translation, that produces an equivalent PDDL+ problem $\Pi_{\text{FLAT}} = \langle F, X_\delta, I_\delta, G, A_\delta, E_\delta, P_\delta \rangle$, whose components are presented in Figure 2. Equation (1) augments the set of numeric predicates $X$ with new fluents, producing $X_\delta$. The fluent ck represents the clock of the system, which keeps track of the flowing of time. Two fluents $\delta_j$ and $\texttt{tk}_j$ are inserted for every partition induced by $J$: the fluent $\delta_j$ keeps track of the value of the discretisation step of the actions of the partition $j$ during the plan and $\texttt{tk}_j$ keeps track of the next time an action will be applicable in that partition (tk stands for *tick*). Equation (2) expands the initial state of the original problem with (i) the set given by $\nabla(I)$, which states the initial value of $\delta^m$, and (ii) the initialisation of the clock and all the ticks of the system to zero. Equation (3) redefines every original action $a$ of $\Pi$, augmenting its precondition with a condition enforcing the action to be applicable only when the clock reaches the correct point, established by the value of $\texttt{tk}_{J(a)}$; here $J(a)$ returns the index of the partition containing $a$. Both in Equation (3) and Equation (4), the effects set of an action or an event $h$ is augmented with the set $u(h)$, defined in Equation (5), in which (i) the value of $\delta_{J(h)}$ is changed to its respective value defined by $\nabla(h)$ only if its value is different from the *persist* value $\kappa$, and (ii) the value of $\texttt{tk}_{J(h)}$ is reset to realign the ticks with the correct value of $\delta_{J(h)}$ set by $\nabla(h)$ (this is because events are not constrained by the ticks). Equation (4) also adds $n$ novel events $\texttt{tic}_j$, defined in Equation (6), which represent the metronome of the system: the event $\texttt{tic}_j$ is fired when the value of the clock ck has just surpassed the value of $\texttt{tk}_j$ by the *simulation delta* of

$$X_\delta = X \cup \{\mathtt{ck}\} \cup \bigcup_{j=1}^{m} \{\delta_j, \mathtt{tk}_j\} \qquad (1)$$

$$I_\delta = I \cup \nabla(I) \cup \{\langle \mathtt{ck} := 0\rangle\} \cup \bigcup_{j=1}^{m} \{\langle \mathtt{tk}_j := 0\rangle\} \qquad (2)$$

$$A_\delta = \bigcup_{a \in A} \{\langle pre(a) \wedge \langle \mathtt{ck} = \mathtt{tk}_{J(a)}\rangle, \mathit{eff}(a) \cup u(a)\rangle\} \qquad (3)$$

$$E_\delta = \bigcup_{\varepsilon \in E} \{\langle pre(\varepsilon), \mathit{eff}(\varepsilon) \cup u(\varepsilon)\rangle\} \cup \bigcup_{j=1}^{m} \{\mathtt{tic}_j\} \qquad (4)$$

$$u(h) = \begin{cases} \emptyset \text{ if } \nabla(h) = \kappa \\ \{\langle \delta_{J(h)} := \nabla(h)\rangle, \langle \mathtt{tk}_{J(h)} := \mathtt{ck}\rangle\} \text{ otherw.} \end{cases} \qquad (5)$$

$$\mathtt{tic}_j = \langle\langle \mathtt{ck} = \mathtt{tk}_j + \delta_e\rangle, \{\langle \mathtt{tk}_j := \mathtt{ck} + \delta_j - \delta_e\rangle\}\rangle \qquad (6)$$

$$P_\delta = P \cup \{\mathtt{t}\} \qquad (7)$$

$$\mathtt{t} = \langle \bigvee_{p \in P} pre(p), \{\langle inc, \mathtt{ck}, \delta_e\rangle\}\rangle \qquad (8)$$

Figure 2: Components of the $\Pi_{\mathrm{FLAT}}$ PDDL+ problem

the planner $\delta_e$ (i.e., we are in the *falling edge*) and, in the effects, it sets the timing ($\mathtt{tk}_j$) in which the raising edge will happen again, taking into consideration the already passed *simulation delta*. Finally, in Equations (7) and (8), a new process $\mathtt{t}$ is added, whose job is to increase the value of the clock $\mathtt{ck}$ by the *simulation delta* $\delta_e$.

It is worth noting that the FLAT translation yielding $\Pi_{\mathrm{FLAT}}$ is polynomial on the size of $\langle \Pi, K_\delta\rangle$. Specifically, FLAT introduces $2m + 1$ numerical variables ($\delta_j$, $\mathtt{tk}_k$ for each $j \in \{1, ..., m\}$ and $\mathtt{ck}$), $m$ events $\mathtt{tic}_j$, where $m$ is the number of partitions of $A \cup E$ induced by $J$, and a single process, i.e., $\mathtt{t}$. Additionally, the preconditions and effects of actions/events are extended with at most 2 numeric conditions and effects. Also, it is easy to see that FLAT preserves the length of a plan exactly; as highlighted by (Nebel 2000), this is a desired property when we talk about compilation from one planning problem into another.

**Theorem 1** (Soundness and Completeness of FLAT w.r.t. $\langle \Pi, K_\delta\rangle$). *Let $\langle \Pi, K_\delta\rangle$ be a PDDL$\delta$+ problem and let $\Pi_{\mathrm{FLAT}}$ be the PDDL+ obtained by using FLAT. $\langle \Pi, K_\delta\rangle$ admits a solution iff $\Pi_{\mathrm{FLAT}}$ does so.*

*Proof Sketch.* ($\Rightarrow$) Let $\pi_t = \langle \pi, t_e\rangle$ be a valid plan for $\langle \Pi, K_\delta\rangle$, and let $\pi'_t = \langle \pi_{\mathrm{FLAT}}, t_e\rangle$ be the plan for $\Pi_{\mathrm{FLAT}}$ constructed in such a way that: for each $i$-th time-stamped action $\langle a_i, t_i\rangle$ in $\pi$, there exists an $i$-th time-stamped action $\langle a'_i, t_i\rangle$ in $\pi_{\mathrm{FLAT}}$, where $a'_i \in A_\delta$ is the action $a_i \in A$ extended with the preconditions and effects introduced by FLAT.

To prove that $\pi'_t$ is a valid plan for $\langle \Pi, K_\delta\rangle$, we approach the proof modularly. Firstly, we note that the problem $\Pi_{\mathrm{FLAT}}$ is an extended version of $\Pi$. Therefore, FLAT does not affect the original part of the problem and $\pi'_t$ achieves $G$.

The important part to prove is that the actions generated by the mapping above are applicable w.r.t. the novel variables. Let $\mathbb{H}$ and $\mathbb{H}'$ be the plan projections generated by $\pi_t$

and $\pi'_t$, respectively. A key element is to prove that the discretisation variables $\delta^m$ evolve in the same way in $\mathbb{H}$ and $\mathbb{H}'$. It is worth noting that in the case of $\mathbb{H}$, the assignments of $\delta^m$ are explicitly kept within $\mathbb{H}$, whereas in $\mathbb{H}'$, they are variables that are part of the problem $\Pi_{\mathrm{FLAT}}$. Specifically, the $\delta^m$ variables of $\mathbb{H}_K$ change when an action (event) is applied (triggered) according to the updating rules R1-R2 of Definition 3, and remain unchanged in other cases. Similarly, the $\delta^m \subset X_\delta$ variables of $\Pi_{\mathrm{FLAT}}$ change whenever an action (event) from $A_\delta$ ($E_\delta$) is applied (triggered). Given the definitions of R1 and R2, along with the actions $A_\delta$ and events $E_\delta$, it is evident that the $\delta^m$ variables are synchronised across all STPs in both $\mathbb{H}$ and $\mathbb{H}'$.

Now, it remains to prove that the actions of $\pi_{\mathrm{FLAT}}$ are applicable. Since FLAT does not affect $F \cup X$ of $\Pi$, the proof focuses only on the new variables $\{\mathtt{ck}\} \cup \bigcup_{j=1}^{n} \{\delta_j, \mathtt{tk}_j\}$. Additionally, it is important to note that for a given partition $A_j$ of $A$, each compiled action from $A_j$ only affects and is affected by $\mathtt{tk}_j$ and $\delta_j$. Therefore, since the partitions of actions of $A_\delta$ do not affect each other, we build the proof by examining a single partition and then generalise the result. So, for a given partition $j$, let $T_K^j = \langle T_1, ..., T_{n_j}\rangle$ ($T_{\mathrm{FLAT}}^j = \langle T'_1, ..., T'_{n_j}\rangle$) be the sequence of STPs in $\mathbb{H}$ ($\mathbb{H}'$) associated with the application of the $n_j$ actions from the partition $j$. We prove by induction that the actions applied in $T_{\mathrm{FLAT}}^j$ are applicable. The *case base* ($i = 1$) trivially proves if $t_1 = 0$. If $t_1 > 0$, we leverage that, (i) $I_\delta \models \langle \mathtt{tk}_j = 0\rangle \wedge \langle \mathtt{ck} = 0\rangle$, (ii) $\mathbb{H}_K(T_1)[M_j] = \mathbb{H}_K(T_0)[M_j] = \langle 0, \delta_j^0\rangle$ and then $t_1 = s \cdot \delta_j^0$ and (iii) for each $T'_0 \leq T' \leq T'_1$, $\mathbb{H}'_s(T')[\delta_j] = \delta_j^0$. Combining these conditions, we obtain that in $[0, t_1]$ the event $\mathtt{tic}_j$ is triggered $s = t_1/\delta_j^0$ times in the STPs $T' = \langle z \cdot \delta_j^0 + \delta_e\rangle$, where $z \in \{0, ..., s - 1\}$. When $z = s - 1$ and $\mathtt{ck} = (s - 1) \cdot \delta_j^0 + \delta_e$, $\mathtt{tic}_j$ sets $\mathtt{tk}_j = \mathtt{ck} + \delta_j^0 - \delta_e = s \cdot \delta_j^0$. Such a value persists until $t_1$ is reached, so that $\mathbb{H}'_s(T'_1) \models \langle \mathtt{ck} = \mathtt{tk}_j\rangle \models pre(a'_i)$. For the induction step, we assume truly the statement for some $1 < i < n_j$ and prove this for $i + 1$. If $t_{i+1} = t_i$, it is easy to see that $a'_{i+1}$ is applicable if $a'_i$ is too (inductive hypothesis). If $t_{i+1} > t_i$, we leverage that, (i) $\mathbb{H}'_s(T'_i) \models pre(a'_i) \models \langle \mathtt{ck} = \mathtt{tk}_i\rangle$ (inductive hypothesis), (ii) $\mathbb{H}'_K(T'_i) = \langle t_i, \nabla(a_i) = \delta_j^i\rangle$ and (iii) for each $T'_i < T' \leq T'_{i+1}$, $\mathbb{H}'_s(T')[\delta_j] = \delta_j^i$. Similarly to the case base, combining these conditions, we obtain that in $[t_i, t_{i+1}]$ the event $\mathtt{tic}_j$ is triggered $s = (t_{i+1} - t_i)/\delta_j^i$ times in the STPs $T' = \langle t_i + z \cdot \delta_j + \delta_e, n'\rangle$, where $z \in \{0, ..., s - 1\}$, thus obtaining a state $\mathbb{H}'_s(T_{i+1}) \models \langle \mathtt{ck} = \mathtt{tk}_j\rangle \models pre(a'_{i+1})$.

($\Leftarrow$) We can proceed in the opposite direction, and hence observing that, starting from a valid plan $\pi'_t = \langle \pi', \langle 0, t_e\rangle\rangle$ for $\Pi_{\mathrm{FLAT}}$, we can create a valid plan $\pi_t = \langle \pi, \langle 0, t_e\rangle\rangle$ for $\langle \Pi, K_\delta\rangle$. The validity of $\pi_t$ can be deduced by the validity of $\pi'_t$ and in particular by noting that each action in $\pi'_t$ implies that the corresponding action is applicable in $\pi_t$ therein. $\square$

Since $\Pi_{\mathrm{FLAT}}$ is an extension of $\Pi$, which does not alter the original problem, it is easy to see that any solution for $\Pi_{\mathrm{FLAT}}$ is also a solution for $\Pi$. However, the converse is not true, as

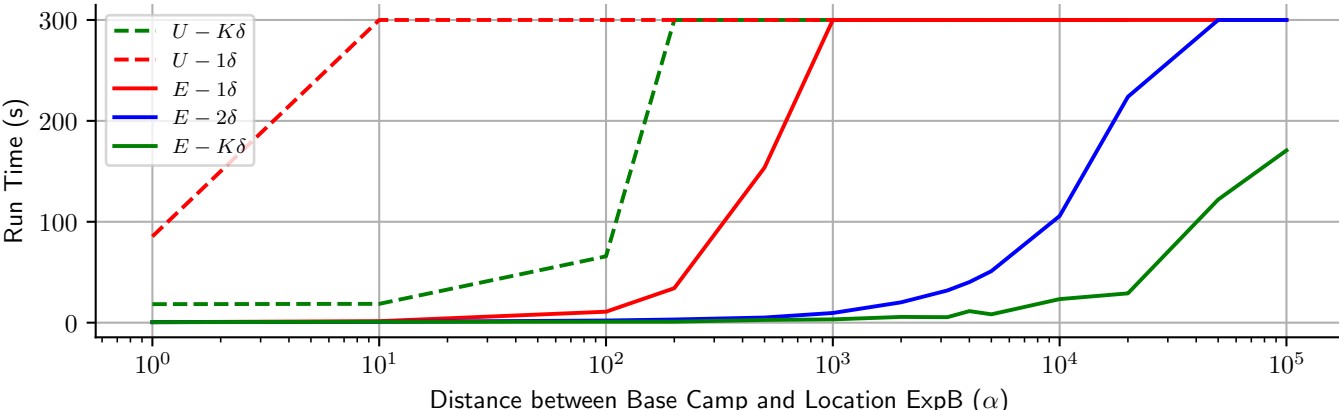

Figure 3: Average runtime (CPU-time seconds) achieved by search approaches implemented in ENHSP (E) and UPMURPHI (U) while relying on different discretisation approaches on the COOPROVERS benchmarks.

all plans with timed actions that are incompatible with $K_\delta$ do not admit a corresponding valid one in $\Pi_{\text{FLAT}}$.

**Corollary 1** (Soundness of FLAT w.r.t. $\Pi$). *Let $\langle \Pi, K_\delta \rangle$ be a PDDL$\delta$+ problem and let $\Pi_{\text{FLAT}}$ be the PDDL+ obtained by using FLAT. $\Pi_{\text{FLAT}}$ admits a solution if $\Pi$ does so.*

## Experimental Analysis

Our experimental analysis aims at assessing how the proposed encoding can affect the performance of PDDL+ domain-independent planning engines. This is done by considering two sets of benchmarks. The first set focuses on the COOPROVERS, where the use of the proposed encoding is expected to deliver a significant performance boost due to the characteristics of the domain. The second set of experiments is performed on well-known PDDL+ benchmark domains with the aim of analysing how our approach behaves in the *Unitary-Static* setting (i.e., only one constant discretisation step). All benchmarks are available at https://anonymous.4open.science/r/deltaExperiments-D585.

Due to the nature of the proposed approach, we focus on planning engines that leverage on discretisation to solve PDDL+ problems. Therefore, we consider two state-of-the-art domain-independent planning engines: ENHSP (Scala et al. 2016) and UPMURPHI (Della Penna, Magazzeni, and Mercorio 2012). ENHSP incorporates a range of heuristics and search techniques, hence providing the ideal ground to compare them within the same infrastructure. UPMURPHI shed some lights into how a radically different approach to discretised PDDL+ can be affected by the proposed translations. In our analysis, we used the default $A^*$ search paired with the default *AIBR* heuristic (Scala et al. 2016), the *add* heuristic (Scala, Haslum, and Thiébaux 2016), and a traditional blind search. UPMURPHI is based on the planning via model-checking paradigm. It automatically translates discretised PDDL+ to a model-checking formulation, and then uses blind search to find a solution. Experiments are run on a 2.3 GHz Intel Xeon 6140M, with a 300 CPU-time seconds cut-off time, and 8 GB RAM.

### Multiple-Dynamic

In the COOPROVERS domain model, choosing the right discretisation step is critical to efficiently generate a valid plan: larger discretisation can lead to draining the battery of the fastest robot. Indeed, to plan faster when the value of $\alpha$ becomes larger, one may consider using a larger discretisation step, proportional to $\alpha$. But this approach can be problematic, as the nearest and fastest robot may not have the possibility to charge before its battery is drained. On the other hand, the use of a smaller discretisation step makes the search space deeper and requires more resources to be explored. Figure 3 shows the results achieved by the considered planning engines when a range of discretisation options are exploited: $1\delta$, the traditional approach in which there is a unique discretisation step to model the granularity of the agent and the environment, i.e., $\delta_e = \delta_p = 1$; $2\delta$, the approach where environment and agent are natively decoupled by the planner by using $\delta_e = 1$ and $\delta_p = 3$ (available only in ENHSP), and the proposed $K_\delta$ approach. The $K_\delta$ approach is run with $\delta_e = 1$ over the PDDL+ problems obtained by using the FLAT translation and exploiting the discretisation knowledge provided for the motivating example in the corresponding section. The ENHSP solver has been run with several heuristics (i.e., *hadd*, *aibr*, *hmrp*) and strategies (i.e., $A^*$ and GBFS), and we show in the plot the minimum run time among all these strategies for each $\alpha$. It can be noted by the line chart how the presented approach better deals with the large value of $\alpha$, allowing to always solve faster than the $2\delta$ approach. The performance improvement is more pronounced when a blind search is used, as in the case of UPMURPHI, where the improvements are noticeable already with small values of $\alpha$. The displayed results confirm that the proposed approach can effectively support the reasoning of domain-independent planning engines in cases where dynamics evolving at different speeds are present in a single planning problem. Further, as a by-product of this work, we note that the newly introduced COOPROVERS domain can provide some interesting test-bed for the planning community, to assess aspects of the planning capabilities of domain-independent approaches that were not considered before.

| | | Baxter | | | Descent | | | HVAC | | | LinearCar | | | SolarRover | | |
|---|---|---|---|---|---|---|---|---|---|---|---|---|---|---|---|---|
| | | $1\delta$ | $2\delta$ | $K\delta$ | $1\delta$ | $2\delta$ | $K\delta$ | $1\delta$ | $2\delta$ | $K\delta$ | $1\delta$ | $2\delta$ | $K\delta$ | $1\delta$ | $2\delta$ | $K\delta$ |
| E+AIBR | RT (s) | 99.3 | **8.6** | 19.4 | 29.7 | **1.8** | 5.7 | 278.3 | **153.8** | 172.9 | 255.1 | **119.0** | 131.6 | 300.0 | **245.1** | 287.5 |
| | Cov. % | 73.6 | **100.0** | 94.7 | 5.0 | **100.0** | 100.0 | 10.0 | **65.0** | 55.0 | 15.0 | **64.6** | 60.4 | 0.0 | **20.0** | 5.0 |
| E+HADD | RT (s) | 126.9 | **0.1** | 16.8 | 300.0 | **205.6** | 263.4 | 300.0 | **2.1** | 61.7 | 300.0 | **285.8** | 300.0 | 300.0 | **198.4** | 275.1 |
| | Cov. % | 57.9 | **100.0** | 94.7 | 0.0 | **35.0** | 15.0 | 0.0 | **100.0** | 85.0 | 0.0 | **5.0** | 0.0 | 0.0 | **60.0** | 15.0 |
| E+BLIND | RT (s) | 289.5 | **139.4** | 212.0 | 300.0 | **216.8** | 269.2 | 300.0 | 300.0 | 300.0 | 300.0 | **285.6** | 300.0 | 300.0 | **286.4** | 296.3 |
| | Cov. % | 5.3 | **57.9** | 36.8 | 0.0 | **30.0** | 15.0 | 0.0 | 0.0 | 0.0 | 0.0 | **5.0** | 0.0 | 0.0 | **5.0** | **5.0** |
| U+BLIND | RT (s) | 300.0 | - | **258.7** | 285.9 | - | **20.1** | 300.0 | - | **297.1** | 300.0 | - | **149.5** | 300.0 | - | 300.0 |
| | Cov. % | 0.0 | - | **15.8** | 5.0 | - | **100.0** | 0.0 | - | **5.0** | 0.0 | - | **54.2** | 0.0 | - | 0.0 |

Table 2: Average runtime (RT, CPU-time seconds) and coverage (Cov.) achieved by informed and uninformed search approaches implemented in ENHSP (E) and UPMurphi (U) while relying on different discretisation approaches on well-known benchmark domains. Average runtime (RT) considers unsolved instances as cut-off time (300 seconds).

## Unitary-Static

In these settings, we aim to understand if, on well-known benchmark instance: (i) the proposed approach can improve the performance of general domain-independent planning engines, and (ii) the proposed approach allows achieving performance that are comparable to those of a planning engine natively exploiting a dual discretisation. We consider the well-known benchmark domains of Baxter, Descent, HVAC, LinearCar, and Rover.

Table 2 provides an overview of the results. Every approach is run using $\delta_e = 0.1$; $2\delta$ and $K\delta$ discretise the agent's action with $\delta_p = 1$, the first natively on the planner side and the second via translation by setting $J(h) = 1$ and $\nabla(z) = 1 \, \forall z \in A \cup E$; finally, $1\delta$ employs $\delta_p = 0.1$.

Remarkably, the use of $K\delta$ allows all the considered planning systems to perform significantly better than when the standard $1\delta$ techniques are in use. Often, this is not only reflected in better runtimes, but also in higher coverage. This strongly indicates that, even in domains where a single delta may seem to be appropriate, an intelligent use of multiple deltas can be beneficial; further, our approach can allow any planning engine to directly benefit from it. Finally, the comparison against the $2\delta$ technique implemented in ENHSP shows that the use of the proposed reformulation of PDDL+ does not add a significant computational overhead. Of course, techniques that are encoded in a planning engine lead to better performance, but it is worth reminding that $K\delta$ gives more flexibility and the possibility to tailor the discretisation step for multiple dynamics.

## Discussion

It is well-known that, in general, finding a suitable discretisation for a continuous system is a hard and challenging task (Della Penna, Magazzeni, and Mercorio 2012). On the one hand, a finer discretisation leads to a more accurate approximation of the continuous behaviour. On the other hand, a more coarse discretisation reduces the size of the search space and fosters solvability. This problem is, of course, exacerbated in approaches where multiple deltas need to be set, i.e. ENHSP or the solution proposed in this paper. However, it is worth noting that in many practical cases, it is easy to find suitable discretisation values, that can be implied by system constraints or by domain knowledge. When multiple agents or systems need to interact, an analysis of the greatest common divisor and of the minimum common multiple among considered delta values for the agents can shed some light into promising values to be used to ensure a good approximation of the interaction points among agents. In extreme cases, where the application domain or the characteristics of the problem at hand do not easily allow to identify suitable discretisation values, the methodology proposed by Della Penna, Magazzeni, and Mercorio (2012) is to start from a coarse discretisation and refine it until the approximation error is within an acceptable threshold. The overall plan-validate framework can support this approach, and has been extensively exploited in real-world applications of hybrid planning (Percassi, Scala, and Vallati 2023a).

## Conclusion

Discretisation is a well-established approach to reason upon challenging hybrid PDDL+ problems. The vast majority of existing approaches are based on a single discretisation step, and ENHSP is the only approach that can leverage two different discretisation steps in a domain-independent fashion. With the aim of taming complex PDDL+ problems where multiple deltas are needed to efficiently generate solutions, in this paper we presented a reformulation approach that allows any domain-independent planning engine to exploit multiple discretisation steps. The formalised notion also allowed us to categorise different levels of discretisation control. The performed experimental analysis highlights the benefits of the discretisation knowledge $K_\delta$ in problems characterised by the coexistence of different dynamics, and also shows the capabilities of the approach on well-known PDDL+ benchmark domains. Our experimental analysis also indicates that the existing benchmarks for PDDL+ lack of variety in terms of modelled dynamics; our motivating example fills this gap, and the proposed approach can equip existing planning engines with the means to solve this class of hybrid planning problems.

As future work, we are interested in investigating approaches that can cover the whole range of discretisation control levels shown in Table 1. We also plan to explore the synergies that can be generated between multiple discretisation reformulations and domain-independent heuristics, to design models that can generate search spaces easier to be navigated by planning engines.

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
