# OpenReview forum: "Taming Discretised PDDL+ through Multiple Discretisations"
_icaps-conference.org/ICAPS/2024/Conference — ICAPS 2024_

### Official Review · Reviewer_ZaUo · 2024-01-22

**Significance And Importance:** 3
**Soundness:** 3
**Novelty:** 3
**Clarity:** 3
**Overall Evaluation:** 2
**Confidence:** 5

**Weaknesses:**

0: Minor weaknesses requiring some work to be addressed for the paper to be accepted.

**Contributions Of The Paper:**

The idea behind this paper is that planning for hybrid discrete continuous planning problems, modelled in PDDL+, can be approached by discretisation of the continuous space. However, the best granularity for the discretisation depends on the scale of the dynamics it describes and, in problems with mixed scales, the best approach is to allow different discretisations to govern different dynamics. The approach encodes the discretisation constants directly within the problem formulation.

The authors adopt PDDL+ as their modelling formalism, but, surprisingly, choose a semantics that is not the one proposed in the paper that introduced PDDL+. Instead, they adopt a semantics based on work by Shin and Davis, which uses a 2D model of time: one dimension is the familiar continuous valued time and the other is a natural number index that orders happenings at the same continuous dimension value of time. The main difference between this approach and the approach taken by Fox and Long is that Fox and Long do not allow actions that interfere with one another to occur at the same time, while the Shin and Davis do allow it, by separating them using the second dimension. The authors do not comment on this difference, which has some bearing in the context of this work. It would be better if they were to explicitly call out the important difference which means that some of the plans they would consider valid are not valid under the PDDL+ semantics defined by Fox and Long.

An example of the subtleties this difference introduces can be seen in this problem:

Action A:
effect: P and t1 = 0

Process X
while P, increase t1 at rate 1

Action B:
precondition: P and t2 = 0
Effect: Q

Process Y
while Q, increase t2 at rate 1

Event C:
when t1 = 1: not P, t1 = 0

Event D:
when t2 = 1: t2 = 0, not Q and if P and t1 = 1 then G

The initial state has t1 and t2 = 0, nothing else true and the goal is G. Is it achievable? Note that A must be applied to allow B to be applied and both of them are required to get the clock processes X and Y to start. Then 1 unit of time later, the preconditions of events C and D are both true. However, the ordering matters and (as the authors note in a footnote) the model is considered broken, because the events interact. However, if we make these events into actions, then the Shin and Davis semantics allows us to specify that they occur in a particular order, D then C. Fox and Long, on the other hand, would not allow A and B to be applied simultaneously (A has to happen strictly before B) and then C must be before D, as events or actions, so there is no way to get the goal G. This is not an indication that the approach that the authors take is problematic, but rather that they must be explicit in acknowledging the difference in the semantics and its implications.

The authors' idea is that a discretisation variable can be associated with partitions of actions and events, so that each partition uses its own variable, and that variable can be updated as the plan advances, so that each stream of activity can follow a distinct discretisation over the timeline. The management of these variables can then be directly captured in the models themselves, so that the plan itself drives the discretisation of the timeline for the next stages of the plan. This is a neat idea. With the exception of the details that arise from the discussion of semantics above, I think this works pretty well.

It is not clear to me whether the extension of the condition observed in the footnote mentioned above, to cover temporal intervals created by the persistance values manipulated in the discretisation leads to valid plans being excluded (I think it can, depending on whether the discretisation has been made fine grained enough to make it possible to discern that two proximal events are actually separated). I don't believe this question is addressed - correct me if I am mistaken. I do like the idea that events can be managed on a finer grained time line than the activity of the agents for which the plan is constructed, but I think that there could be challenges when the events are triggered by the actions of the agents, since that introduces a "lumpiness" into the timelines for the events that might make the environmental behaviour pathological when that could have been avoided by allowing the agents to initiate activity at times more closely synchronised with specific points along the timeline.

I am suspicious of the completeness claim and the "proof" is not sufficient to convince me.

**Ethical Considerations:**

(1) Not Applicable: The paper does not have any ethical considerations to address

**Nomination For Best Paper:**

No

**Questions For Authors:**

Can you comment on the question of whether mixed discretisation can lead to some plans being excluded because of the requirement that events must commute? I believe that the imposition of a discretisation is an added constraint that can make some plans no longer possible. How is that addressed in the completeness claim?

**Reproducibility:**

4: Authors promise to release code and domains (whichever apply).

**Strengths Of The Paper:**

The implementation demonstrates the value of the approach and I think that is a strength of the work. Despite my issues with the theoretical details, the implementation demonstrates practical value and I think that this is a useful contribution. I urge the authors to return to the theory and see whether they can tighten up the details.

**Weaknesses Of The Paper:**

The comments above describe the issues I have.

---

> ### Author Rebuttal · Authors · 2024-01-26
>
> We thank the reviewer for the interest in our paper and the insightful review.
>
> Concerning the two different semantics, we took the formulation directly from Percassi, F., Scala, E., \& Vallati, M. (JAIR, 2023). In the paper, at the end of page 120, a discussion about the difference between the two semantics, as underlined by the reviewer, is made explicit. We will clarify this in the paper and refer the reader to the existing discussion for further insights.
>
> The reviewer is correct. The introduction of our delta knowledge can indeed prevent some plans happening. The constraints are indeed pruning some solution. Note, however, that Theorem 1 shows that the presented encoding is sound and complete w.r.t. the problem with the delta knowledge constraints, i.e. PDDL$\delta$+ $\langle \Pi, K_\delta \rangle$, not the overall problem, i.e. $\Pi$, with which it only is sound (Corollary 1). Note also that the delta knowledge does not change the time points in which events and processes may be triggered, but of course, disallowing actions in those time points which are not enabled by the associated discretisation, we may miss some opportunity to find a shorter plan.

---

### Official Review · Reviewer_Tu99 · 2024-01-22

**Significance And Importance:** 2
**Soundness:** 4
**Novelty:** 2
**Clarity:** 3
**Confidence:** 5

**Weaknesses:**

0: Minor weaknesses requiring some work to be addressed for the paper to be accepted.

**Contributions Of The Paper:**

The paper describes a planner-independent extension to PDDL+ planning that allows the use of multiple time discretizations in a single planning problem. Rather than modifying the planner to accept multiple discretizations, the authors propose a new multi-discretized PDDL+ problem which natively incorporates different discretization intervals. A new domain, CoOpRovers, is introduced in the paper to showcase the disadvantages of typical uniform discretization approaches to PDDL+ planning and to motivate the need for dynamic discretization methods. Further experimental evaluation on standard PDDL+ benchmark domains shows the benefits of multiple discretizations compared to the conventional static uniform discretization.

**Ethical Considerations:**

(1) Not Applicable: The paper does not have any ethical considerations to address

**Nomination For Best Paper:**

No

**Overall Evaluation:**

-1: (weak reject)

**Questions For Authors:**

1. is there a pre-processing script that can transform an existing PDDL+ problem into a dynamic planning-discretization PDDL+ problem?
2. Are all the multi-discretization parameters selected manually per domain/problem?
3. What is the concrete implementation of the Del function controlling the dynamics of the discretization steps? Does it need to be explicitly defined for each domain or problem? How can the function infer discretization steps from both the set of actions and events as well as the initial state?
4. Are all the discretization variables multiples of one of the smallest-valued discretization quantum?
5. PDDL+ models are supposed to be continuous by default, could the dynamic planning-discretized PDDL+ problem be (in principle) solved by non-discretization-based planners (e.g., SMTPlan+)?

**Reproducibility:**

5: Code and domains (whichever apply) are already publicly available

**Strengths Of The Paper:**

The paper is well written and motivates the problem quite well. It addresses an important issue in PDDL+ planning that is rather understudied. The main strengths of the approach is that it is a clear improvement over single uniform discretization and does not require modifying the planner. The results are promising when compared with static uniform discretization approaches. The soundness and completeness analyses are welcome additions.

**Weaknesses Of The Paper:**

The main idea is somewhat simplistic and the results show that the proposed approach (K_\delta) is comprehensively outperformed by the double discretization (2\delta) adopted bu ENHSP.
Further, the paper does not specify how the values of \delta discretizations or other important parameters are selected. In its current form, the approach does seem like a handcrafted modification to existing PDDL+ domains which limits its applicability to solving novel PDDL+ problems. Having to directly modify the PDDL+ models and being outperformed by out-of-the-box ENHSP significantly limits the applicability of the proposed approach.

---

> ### Author Rebuttal · Authors · 2024-01-26
>
> We thank the reviewer for the insightful comments and suggestions.
>
> In the experimental analysis we showed that $K\delta$ outperforms $2\delta$ when the domain is in great need of a differentiation in the discretisation, i.e. with the CoOpRovers domain in Figure 3. With Table 2, we showed that $K\delta$ outperforms $1\delta$, which is the standard for UPMurphi, thus allowing the planner to benefit from the approach for free. Moreover, $K\delta$ is not so distant from $2\delta$ which is only employed by ENHSP, and which, being a native approach inside the planner, is advantaged but less general.
>
> 1. Yes, we will make it available in a dedicated repository.
>
> 2. Yes, As done at the state of the art for the discretisation values of planning engines such as ENHSP or UPMurphi. In future work, we would like to investigate how $K\delta$, or a set of suitable values, can be automatically inferred from $\Pi$.
>
> 3. The $\nabla$ function is specified in a JSON as a simple dictionary whose keys can either be lifted or ground action names and as value has an array of floats of length $m$, denoting all the $\delta_1, \dots, \delta_m$. A special key of the dictionary sets the initial condition.
>
> 4. Yes, they must be all multiple of the discretisation value $\delta_e$.
>
>
> 5. It is in principle possible to run a continuous time planner over our encoding. Yet, our theoretical claims hold for the discrete case, and we have not investigated what happens under a continuous interpretation of time. We can speculate that a continuous time PDDL+ based on SMT such as SMTPlan+ won't benefit from our approach as our encoding greatly increases the number of significant time points (and this would translate in very large formulas); rather one would need to think of a more direct way to support the delta knowledge constraints. Thanks for the question, we will look into this and include the results in the CRC if allowed.

---

### Official Review · Reviewer_5whB · 2024-01-23

**Significance And Importance:** 2
**Soundness:** 3
**Novelty:** 3
**Clarity:** 4
**Overall Evaluation:** 1
**Confidence:** 4

**Weaknesses:**

1: Minor weaknesses that are easily fixable.

**Contributions Of The Paper:**

The paper under review addresses the challenging task of solving PDDL+ hybrid domains, where the incorporation of discrete-continuous variables and processes adds complexity to the problem. The central idea revolves around the formalization of a dynamic planning PDDL+ problem that encompasses the concept of variable discretization. While the notion of using variable discretization is not new, as previously proposed by Fox et al in "Automatic Construction of Efficient Multiple Battery Usage Policies" (ICAPS-11), the paper introduces a reformulation that enables any existing PDDL+ planner to reason on the reformulated models. This significantly contributes to the effective handling of complex hybrid domains. The paper is well-written, the formalization is robust, and the SOTA coverage is comprehensive.

**Ethical Considerations:**

(1) Not Applicable: The paper does not have any ethical considerations to address

**Nomination For Best Paper:**

No

**Questions For Authors:**

1. The authors argue that the appropriate discretization steps can often be derived from domain constraints, but it is unclear how or by whom the discretization step is set. I suggest dynamically adjusting the step during plan execution based on the partial plan. The authors are encouraged to comment on this aspect for clarity.
2. Considering that reasoning with discretized domains and planning may necessitate plan validation, it is suggested that the authors address using VAL to validate generated plans in their methodology.
3. The approach presented in the paper may benefit the DiNo planner; however, it is not included in the experimental results. The authors are encouraged to motivate the exclusion and discuss how their approach could benefit the DiNo planner.

**Reproducibility:**

5: Code and domains (whichever apply) are already publicly available

**Strengths Of The Paper:**

1. Formalization enables existing state-of-the-art PDDL+ planners to handle variable discretization steps.
2. Introduction of a novel benchmark PDDL+ domain, COOPROVERS.
3. Valuable contribution to the community for more effective handling of complex PDDL+ mixed discrete-continuous domains.

**Weaknesses Of The Paper:**

see questions

---

> ### Author Rebuttal · Authors · 2024-01-26
>
> We thank the reviewer for the appreciation of our work.
>
> 1. On the choice of discretisation: presently we focus on the translation approach on problems, like CoOpRovers, where the delta knowledge is part of the problem and can be set a priori, like in the $2\delta$ approach of ENHSP. In this work, we wanted to present a planner-independent approach. In future work, we would like to explore the possibility to adjust the discretisation while planning in a dynamic way.
>
> 2 and 3. On the use of VAL and DiNo: Using a discrete semantic, we couldn't use VAL for validation. The internal validator of ENHSP was used instead.
> DiNo is built on top of the UPMurphi framework in fact, adding a heuristic on UPMurphi, which is included in the analysis. Further, DiNo did not find any solution to the considered benchmarks in the given time limit, and hence we excluded it from the analysis. We will clarify this.

---

### Meta-Review · Area_Chair_k9Yj · 2024-02-06

**Recommendation:** Accept (Poster)
**Confidence:** 5

**Metareview:**

We recommend to accept this paper. While there is not uniformity of opinion among reviewers according to headline scores, there is agreement that the work can be accepted, and recognition of the need to drive work in PDDL+ planning towards being suitable for practical use.

**Ethical Considerations:**

(1) Not Applicable: The paper does not have any ethical considerations to address